



# Fractional snow-covered area: Scale-independent peak of winter parameterization

Nora Helbig [1], Yves Bühler [1], Lucie Eberhard [1], César Deschamps-Berger [2,3], Simon Gascoin [2], Marie Dumont [3], Jesus Revuelto [3,4], Jeff S. Deems [5], and Tobias Jonas [1]

[1]WSL Institute for Snow and Avalanche Research SLF, Davos, Switzerland
[2]Centre d'Etudes Spatiales de la Biosphère, UPS/CNRS/IRD/INRAE/CNES, Toulouse, France
[3]Univ. Grenoble Alpes, Université de Toulouse, Météo-France, CNRS, CNRM, Centre d'Études de la Neige, 38000 Grenoble, France
[4]Instituto Pirenaico de Ecología, Consejo Superior de Investigaciones Científicas (IPE–CSIC), Zaragoza, Spain
[5]National Snow and Ice Data Center, University of Colorado, Boulder, CO, USA

**Correspondence:** Nora Helbig (norahelbig@gmail.com)

**Abstract.** The spatial distribution of snow in the mountains is significantly influenced through interactions of topography with wind, precipitation, shortwave and longwave radiation, and avalanches that may relocate the accumulated snow. One of the most crucial model parameters for various applications such as weather forecasts, climate predictions and in hydrological modeling is the fraction of the ground surface that is covered by snow, also called fractional snow-covered area ($fSCA$). While previous subgrid parameterizations for the spatial snow depth distribution and $fSCA$ work well, performances were scale-dependent. Here, we were able to confirm a previously established empirical relationship of the peak of winter parameterization for the standard deviation of snow depth $\sigma_{HS}$ by evaluating it on 11 spatial snow depth data sets from 7 different geographic regions and snow climates with resolutions ranging from 0.1 m to 3 m. Enhanced performance (mean percentage errors (MPE) decreased by 25 %) across all spatial scales $\geq$ 200 m was achieved by recalibrating and introducing a scale-dependency in the dominant scaling variables. Scale-dependent MPEs vary between -7 % and 3 % for $\sigma_{HS}$ and between 0 % and 1 % for $fSCA$. A scale- as well as region-dependent evaluation revealed that for the majority of the regions the MPEs mostly lie between $\pm 10$ % for $\sigma_{HS}$ and between -1 % and 1.5 % for $fSCA$. This suggests that the new parameterizations perform similarly well in most geographical regions.

## 1 Introduction

Whenever there is snow on the ground, there will be large spatial variability in snow depth. In mountainous terrain, this spatial distribution of snow is significantly influenced by topography due to corresponding spatial variations in wind, precipitation, shortwave and longwave radiation, and in steep terrain by avalanches that may relocate the accumulated snow. As a result, the snow-covered landscape can consist of a complex mix of snow-free and snow-covered areas, especially in steep terrain or during snow melt. A parameter which describes how much of the ground is covered by snow is the fractional snow-covered area ($fSCA$). Most of the time $fSCA$ is tightly linked to snow depth ($HS$) and in particular to its spatial distribution. $fSCA$ is able to bridge between the spatial mean $HS$ and the actual observed snow coverage. Sound $fSCA$ models are therefore



crucial, since for the same mean $HS$ in early winter and in late spring the associated $fSCA$ can be completely different (e.g. Luce et al., 1999; Niu and Yang, 2007; Magand et al., 2014).

$fSCA$ plays a key role in modelling physical processes for various applications such as weather forecasts (e.g. Douville et al., 1995; Doms et al., 2011), climate simulations (e.g. Roesch et al., 2001; Mudryk et al., 2020) and avalanche forecasting (Bellaire and Jamieson, 2013; Horton and Jamieson, 2016; Vionnet et al., 2016). As climate warms, $fSCA$ is an highly relevant indicator for spatial snow cover changes in climate projections (e.g. Mudryk et al., 2020). A decrease in spatial snow extent prominently changes surface characteristics such as albedo in mountain landscapes, leading to changes in surface radiation, a primary component of the surface energy balance. $fSCA$ is also parameter in hydrological models to scale water discharges in the different model grid cells managing in this way appropriately basins water supply (e.g. Luce et al., 1999; Thirel et al., 2013; Magnusson et al., 2014; Griessinger et al., 2016). Errors in $fSCA$ estimates directly translate into errors of snow melt rates and melt water discharge (Magand et al., 2014). Thus, accurately describing $fSCA$ is of key importance for multiple model applications in mountainous terrain where highly variable spatial snow distributions occur.

$fSCA$ can be obtained from satellite remote sensing products using optical imagery with varying spatiotemporal resolution. For instance, Sentinel-2 gathers data at a spatial resolution of 10 to 20 m at frequent global revisit intervals (<5 days, cloud-permitting) (Drusch et al., 2012; Gascoin et al., 2019). The availability of satellite-derived $fSCA$ remains however inconsistent due to time gaps between satellite revisits, data delivery and the frequent presence of clouds, which obscure the ground, especially in winter in mountainous terrain reducing the availability of images drastically (e.g. Parajka and Blöschl, 2006; Gascoin et al., 2015). Satellite-derived $fSCA$ can also not be used directly for forecasting. Alternatively, $fSCA$ can be obtained from spatially averaging by using snow models at subgrid scales. While such snow cover models are available (e.g. Tarboton et al., 1996; Marks et al., 1999; Lehning et al., 2006; Essery et al., 2013; Vionnet et al., 2016), up until now they cannot be used in very high spatial resolutions over very large regions, in part due to a lack of detailed input data, such as fine-scale surface wind speed and precipitation, as well as due to high computational cost. Often they are limited by model parameters and structure requiring calibration. Integrating data assimilation algorithms in snow models is able to mitigate some of these limitations which led for instance to improvements in runoff simulations (e.g. Andreadis and Lettenmaier, 2006; Nagler et al., 2008; Thirel et al., 2013; Griessinger et al., 2016; Huang et al., 2017; Griessinger et al., 2019). However, uncertainties inherently present in the input or assimilation data still remain, which are generally accentuated over snow-covered catchments (Raleigh et al., 2015). Today, $fSCA$ parameterizations describing the subgrid snow depth variability therefore remain unavoidable for complex model systems and to complement assimilation of satellite-retrieved $fSCA$ products especially over mountainous terrain.

A parameterization of $fSCA$ describes the relationship between $fSCA$ and grid cell-averaged $HS$ or snow water equivalent ($SWE$) by a so-called snow-cover depletion ($SCD$) curve. $SCD$ curves were originally introduced in models without taking into account subgrid topography or vegetation. In principle, there are two commonly applied forms: so-called closed functional forms and parametric probabilistic $SCD$ curve formulations (Essery and Pomeroy, 2004). Parametric $SCD$ curves have disadvantages for practical applications such as numerical stability, computational efficiency and assuming an unimodal distribution which might be less appropriate for large grid cells covering heterogeneous surface such as mountainous terrain



(e.g. Essery and Pomeroy, 2004; Swenson and Lawrence, 2012). Various closed functional forms for $fSCA$ are therefore applied in land surface and climate models (e.g. Douville et al., 1995; Roesch et al., 2001; Yang et al., 1997; Niu and Yang, 2007; Su et al., 2008; Swenson and Lawrence, 2012). Most of these parameterizations use simple relationships between $fSCA$
and $HS$ or $SWE$. Since topography strongly determines the spatial $HS$ or $SWE$ distribution (Clark et al., 2011), in the past, terrain characteristics were mostly heuristically introduced in closed form curves to account for subgrid terrain influences on $fSCA$ (e.g. Douville et al., 1995; Roesch et al., 2001; Swenson and Lawrence, 2012). To verify the commonly applied closed forms of $fSCA$, Essery and Pomeroy (2004) integrated over log-normal $SWE$ distributions and fitted the parametric $SCD$ curves. The best obtained fit resulted for a function proportional to $tanh$ which is a previously derived closed form from Yang
et al. (1997). By using a normal probability density function (pdf) Helbig et al. (2015) obtained the same form fit for $fSCA$ as Essery and Pomeroy (2004). The functional form for $fSCA$ from Yang et al. (1997) could thus be inferred from integrating normal as well as log-normal $HS$ distributions with subsequent fitting of the parametric $SCD$ curves. The main difference between the form of Yang et al. (1997) and Essery and Pomeroy (2004) is the variable in the denominator. Yang et al. (1997) used the aerodynamic roughness length whereas Essery and Pomeroy (2004) obtained the standard deviation of snow depth
($\sigma_{HS}$) at peak of winter in the denominator. The advantage of introducing $\sigma_{HS}$ in the $fSCA$ parameterization is that subgrid terrain characteristics, contributing to shape the dominant spatial snow depth distribution, can be used to parameterize $\sigma_{HS}$ (Helbig et al., 2015).

Until recently, it was not possible to derive an empirical parameterization for $\sigma_{HS}$ based on high-resolution $HS$ data due to the lack of high-resolution spatial $HS$ data. New measurement methods such as terrestrial laser scanning (TLS), airborne
laser scanning (ALS) and airborne digital photogrammetry (ADP) nowadays provide a wealth of spatial $HS$ data at fine-scale horizontal resolutions. Since recently, digital photogrammetry can also be applied to high-resolution optical satellite imagery (Marti et al., 2016; Deschamps-Berger et al., 2020; Eberhard et al., 2020; Shaw et al., 2020). $HS$ data at these high resolutions now allow to statistically analyze spatial snow depth patterns for various purposes (e.g. Melvold and Skaugen, 2013; Grünewald et al., 2013; Kirchner et al., 2014; Grünewald et al., 2014; Revuelto et al., 2014; Helbig et al., 2015; Voegeli
et al., 2016; López-Moreno et al., 2017; Helbig and van Herwijnen, 2017; Skaugen and Melvold, 2019). Based on spatial snow depth data sets, $\sigma_{HS}$ could be related to terrain parameters. For instance, Helbig et al. (2015) parameterized $\sigma_{HS}$ at peak of winter using spatial mean $HS$ and subgrid terrain parameters, namely a squared slope related parameter and terrain correlation length, and Skaugen and Melvold (2019) parameterized $\sigma_{HS}$ for the accumulation season using current spatial mean $HS$ and stratifications according to landscape classes and standard deviations of squared slope. Though both approaches
are promising and also somehow similar, e.g. both use the squared slope as significant scale variable, they also differ, e.g. in the considered horizontal scale lengths at the development of the parameterization. While the parameterization of Helbig et al. (2015) was developed for squared grid cell sizes from 50 m to 3 km, Skaugen and Melvold (2019) presented parameterizations for 0.5 km x 1 km grid cells. Helbig et al. (2015) observed improved performances for larger scales ($> 1000$ m), Skaugen and Melvold (2019) observed the same performances when validating it for 0.5 km x 10.25 km grid cells. This can be explained
by the physical processes shaping the complex mountain snow cover predominantly interacting at different length scales with topography e.g. precipitation, wind and radiation (Liston, 2004). A multi-scale behaviour has been found in various studies





using different spatial coverages and measurement platforms (e.g. Deems et al., 2006; Trujillo et al., 2007; Schirmer et al., 2011; Mendoza et al., 2020), but a thorough analysis of spatial autocorrelations using many spatial snow depth data sets up to several kilometers in horizontal resolutions far below the first estimated scale break of about 10 to 20 m has not been presented

so far. Such an analysis could reveal a scale range from which the spatial snow distribution in mountainous terrain can be parameterized with consistent accuracy. Using the newly available wealth of spatial $HS$ data we now have the opportunity to better understand the differences in previous empirically developed closed-form $fSCA$ parameterizations by adding variability in evaluation data sets, i.e. by using data from different geographic regions, as well as by taking into account the spatial scale in scaling parameters.

This article presents a new peak of winter $fSCA$ parameterization for mountainous terrain for various snow model applications. Since snow model applications operate at different spatial scales a $fSCA$ parameterization should work across spatial scales as well as for various snow climates. Two important points were therefore tackled compared to a previous peak of winter $fSCA$ parameterization: 1) We derived the empirical parameterization for $\sigma_{HS}$ on a large pool of spatial snow depth data sets from various geographic sites and validated it scale- as well as region-dependent. 2) Based on a spatial scale analysis

we introduced scale-dependent parameters in the parameterization of Helbig et al. (2015) for $\sigma_{HS}$ such that the new $fSCA$ parameterization is scale-independent for grid cell sizes starting at 200 m up to 5 km.

## 2   Data

We compiled the large quantity of 11 spatial snow depth data sets from seven different geographic sites in mountainous regions of Switzerland, France and the US, i.e. from two continents (Figure 1). These data sets have horizontal grid cell resolutions $\Delta x$

between 0.1 m and 3 m and cover areas from 0.14 km$^2$ to 280 km$^2$. In addition to that, the snow depth data sets were acquired by five different remote sensing methods, i.e. using different platforms. The diversity of the data sets can be seen in Figure 2 showing the pdfs for snow depth, elevation and the squared slope related parameter $\mu$ (Helbig et al., 2015) which is described in Section 3.3. All snow depth data was gathered at the local approximate point in time when snow accumulations had reached their annual maximum. Except for the two snow depth data sets shown in Figure 3, the data sets have been published before or

the geographic location is described elsewhere. In the following all snow depth data sets are listed, grouped according to their mountain range.

### 2.1   Eastern Swiss Alps

We used snow depth data sets acquired by three different platforms above four different alpine sites in the eastern Swiss Alps.

The first platform was airborne digital scanning (ADS) using an opto-electronic line scanner on an airplane. Data was

acquired from the Wannengrat and Dischma area near Davos in the eastern Swiss Alps (Bühler et al., 2015). ADS-derived snow depth data sets were used from 20 March 2012 ('ads-CH$^2$') and 9 March 2016 ('ads-CH$^1$') together with summer digital elevation models (DEM) (Marty et al., 2019). The data set covers about 150 km$^2$ in 2 m resolution. Bühler et al. (2015)


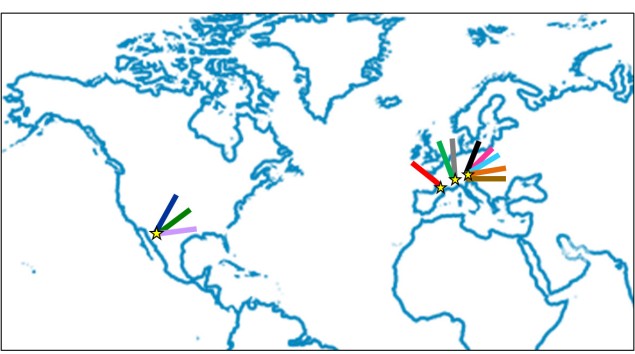

**Figure 1.** The map shows the approximate location of the eleven spatial snow depth data sets. The colors of the trays indicate the region, measurement platform or acquisition date as presented in Figure 2.

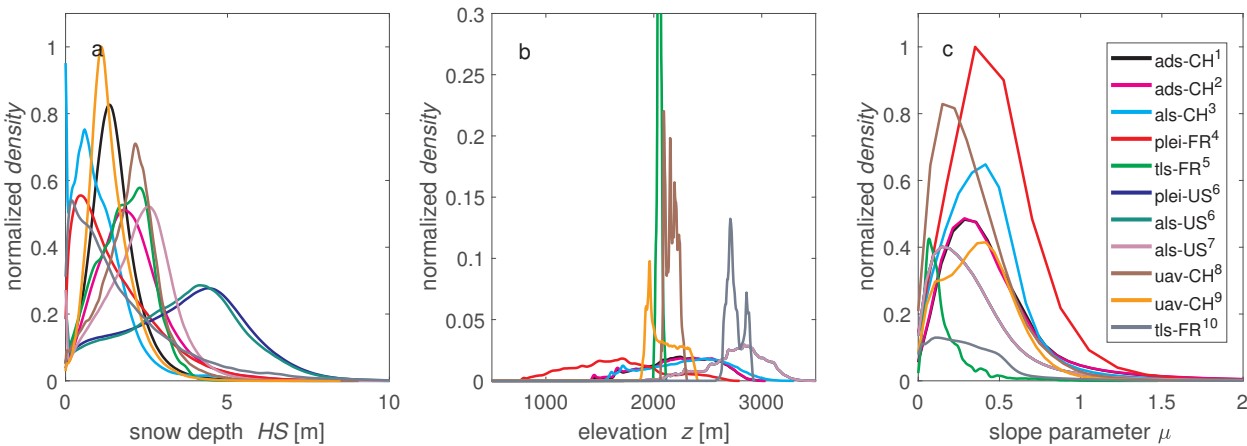

**Figure 2.** Probability density functions for (a) snow depth $HS$, (b) elevation $z$ and (c) squared slope related parameter $\mu$ per data set in its original horizontal resolution, i.e. between 0.1 m and 3 m. All densities were normalized with the maximum of all data sets. Note that for elevation (b) the y-axis was cut for better visibility. Colors represent the different geographic regions, measurement platform or acquisition dates (number) of the compiled data set as indicated in Section 2.1 to 2.4.





validated the 2 m ADS-derived snow depth data among others with TLS data. They obtained a root mean square error (RMSE) of 33 cm and a normalized median absolute deviation (NMAD) of the residuals (Höhle and Höhle, 2009) of 26 cm.

The second platform was an unmanned aerial system (UAS) recording optical imagery with real time kinematik (RTK) positioning of the image acquisition points of the snow cover by a standard camera over two different smaller regions near Davos in the eastern Swiss Alps (Bühler et al., 2016; Eberhard et al., 2020). These images were photogrammetrically processed into a digital surface model DSM. By subtracting the snow free DSM from the summer flight, the HS values were obtained (Bühler et al., 2017). An UAS-derived snow depth data set was used from 7 April 2018 ('uav-CH[9]') from Schürlialp together

with a UAS-acquired summer DEM (Eberhard et al., 2020). The Schürlialp data set covers about 3.2 $km^2$ which we used in 30 cm resolution. A second UAS-derived snow depth data set was used from 29 March 2019 ('uav-CH[8]') from Gaudergrat together with a UAS-acquired summer DEM. The Gaudergrat data set covers about 0.8 $km^2$ in 10 cm resolution (Figure 3b). Compared to snow depth data from snow probing, Eberhard et al. (2020) obtained a RMSE of 16 cm and a NAMD of 11 cm for UAS-derived snow depth data at 9 cm horizontal resolution from Schürlialp.

The third platform was airborne laser scanning (ALS) above the Dischma region near Davos in the eastern Swiss Alps (Figure 3a). This acquisition was a Swiss partner mission of the Airborne Snow Observatory (ASO) (Painter et al., 2016). For consistency reasons, the same lidar setup was used and similar processing standards than for the ASO campaigns in California were applied (Section 2.2). ALS-derived snow depth data was used from 20 March 2017 ('als-CH[3]') together with a summer DEM from 2017. The ALS data set from Switzerland used here covers about 260 $km^2$ in 3 m resolution. Details on the

derivation of the ALS data can be found in Mazzotti et al. (2019) though this study focused on three 0.5 $km^2$ forested sub data sets. Validation of 1 m ALS-derived snow depth grids from 20 March 2017 against data from snow probing within forest but outside canopy (i.e. not below a tree) resulted in a RMSE of 13 cm and a bias of -5 cm.

## 2.2  Sierra Nevada, CA, US

We used data sets acquired by two different platforms above Tuolumne basin in the Sierra Nevada (California) in the US.

The first platform was ALS performed by ASO (Painter et al., 2016). ALS-derived snow depth data was used from 26 March 2016 ('als-US[7]') and 2 May 2017 ('als-US[6]') together with a summer DEM (Painter, 2018). The second platform was a Pléiades product from 1 May 2017 ('plei-US[6]'). A detailed data description of the Pléiades data set derivation is given in Deschamps-Berger et al. (2020).

    We used the ASO summer DEM for the Pléiades as well as the ALS snow depth data sets. Given that the extent of the

Pléiades snow depth data set was much smaller than the ALS domain, we cropped the ALS data sets to the Pléiades data set extension resulting in a coverage of about 280 $km^2$. The horizontal resolution used here was 3 m for both data sets. Compared to snow probe measurements in relatively flat areas ALS snow depth data at 3 m horizontal resolution was found unbiased with a RMSE of 8 cm (Painter et al., 2016). Pléiades-derived snow depth data was recently validated with ASO data over 137 $km^2$ at 3 m resolution above Tuolumne basin (Deschamps-Berger et al., 2020). A RMSE of 80 cm, a NMAD of 69 cm and a mean

bias of 8 cm was obtained for the Pléiades data set.
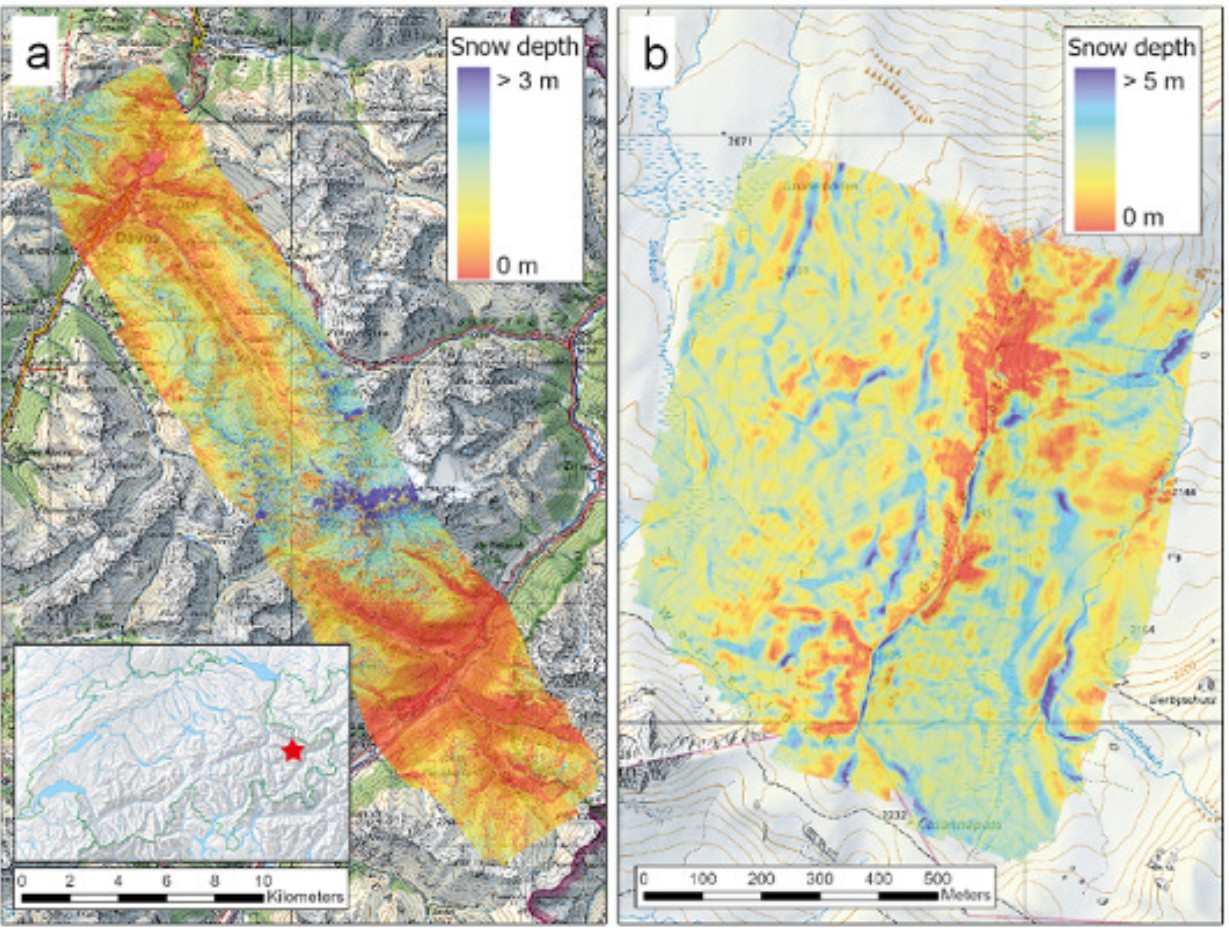

**Figure 3.** Snow depth maps of the eastern Swiss Alps: (a) in the Dischma region (ALS data) and (b) at Gaudergrat (UAS data) at peak of winter. The red dot in the inset map for Switzerland shows the location of the two sites. Pixmap © 2020 Swisstopo (5704000000), reproduced by permission of Swisstopo (JA100118).





### 2.3 Eastern French Pyreenes

A Pléiades product was acquired over the Bassiès basin in the northeastern French Pyreenes. Pléiades-derived snow depth data was used from 15 March 2017 ('plei-FR[4]') together with a summer DEM (Marti et al., 2016). The data set we used, covers about 113 km$^2$ in 3 m resolution. Marti et al. (2016) derived a median of the bias between 2 m Pléiades data and snow probe measurements of -16 cm and with UAS measurements of -14 cm. They further obtained a NMAD of 45 cm with snow probe measurements and a NMAD of 78 cm with UAS measurements.

### 2.4 Southeastern French Alps

TLS-derived snow depth data was acquired at two alpine mountain passes in the southeastern French Alps. One snow depth data set was acquired over Col du Lac Blanc at 9 March 2015 ('tls-FR[10]') (Revuelto et al., 2020). A site and data description can be found in Naaim-Bouvet et al. (2010); Vionnet et al. (2014); Schön et al. (2015, 2018). We used a UAS-acquired summer DEM (Guyomarc'h et al., 2019). The data set covers about 0.6 km$^2$ in 1 m resolution. The second TLS-derived snow depth data set was acquired over Col du Lautaret at 27 March 2018 ('tls-FR[5]') (Revuelto et al., 2020, under review). We used a TLS-acquired summer DEM. The data set covers about 0.14 km$^2$ in 1 m resolution. Previously, mean biases between 4 and 10 cm for TLS laser target distances up to 500 m were obtained between TLS-derived and reference tachymetry measurements (Prokop, 2008; Prokop et al., 2008; Grünewald et al., 2010).

### 2.5 Preprocessing

In all data sets grid cells $\Delta x$ with forest, rivers, glaciers or buildings were masked out. In order to avoid introducing any biases we consistently neglected snow depth values in all data sets that were lower zero or above 15 m. We used a $HS$ threshold of zero to decide whether or not a grid cell was snow-covered.

## 3 Methods

Following the approach of Helbig et al. (2015), we parameterize the standard deviation of snow depth $\sigma_{HS}$ to reassess the validity of the $fSCA$ parameterization for complex topography of Helbig et al. (2015) for a range of spatial scales, in particular for sub-kilometer spatial scales.

### 3.1 Aggregating and pooling of data sets

Pooling all snow depth data sets yields a data pool with a vast variety in snow climates, topographic characteristics and thus snow depth distributions. We first aggregated all snow depth data in squared so-called domain sizes $L$ in regular grids between 3 m to 5 km. Our choice of the smallest applicable $L$ in a data set was defined by a large enough $L/\Delta x$ ratio (here $\geq 20$) to minimize the influence of grid cell resolutions when spatially averaging (Helbig et al., 2009). When aggregating, we required at least 70 % valid data in a domain size which was the maximum threshold to obtain a sufficient number of domains for the



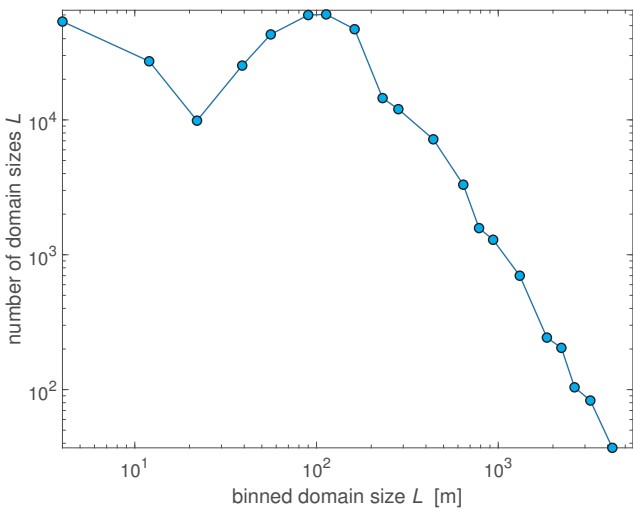

**Figure 4.** Total number of valid domain sizes $L$ per binned $L$ in log-log scale.

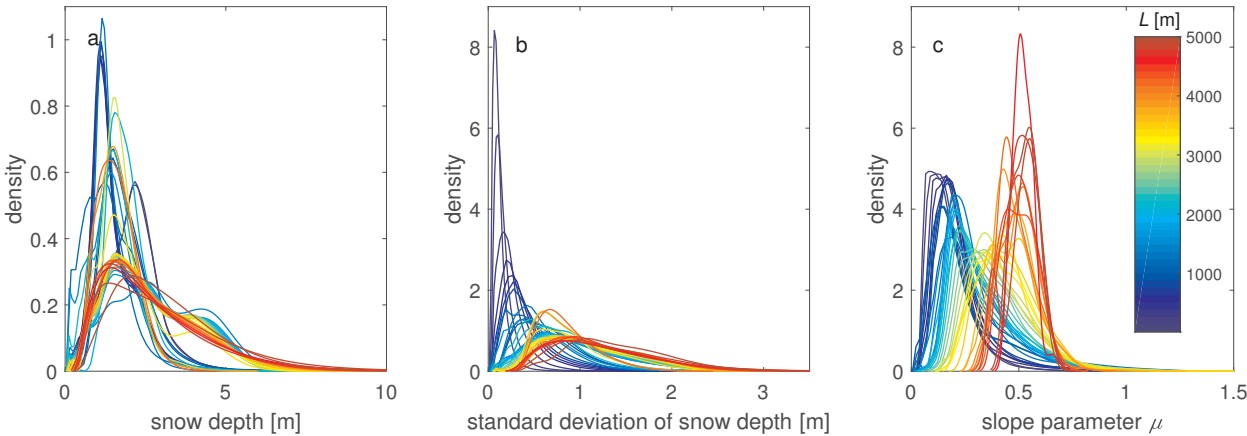

**Figure 5.** Probability density functions for mean $HS$, $\sigma_{HS}$ and squared slope related parameter $\mu$ per domain size $L$ after preprocessing and pooling all data sets.

largest domain sizes $L$ of 3 to 5 km. In addition to that we excluded $L$ with spatial mean slope angles larger than 60 ° and spatial mean snow depth $HS$ lower than 5 cm. By applying these limitations and since horizontal resolutions $\Delta x$ as well as the overall extent of the data sets vary, the full range of $L$ was not represented by each data set. Overall this resulted in a pool of 367'643 domain with $L$ between 3 m and 5 km. We obtain a decreasing number of domains for increasing $L$ with a range between 59'376 for $L$ = 90 m and 17 for $L$ = 5000 m (Figure 4). The diversity of the remaining $L$ is shown by means of the

pdfs for $HS$, $\sigma_{HS}$ and the squared slope related parameter $\mu$ in Figure 5. Spatial averages and standard deviations were built for each $L$. In the following, overbars are neglected for spatial averages.



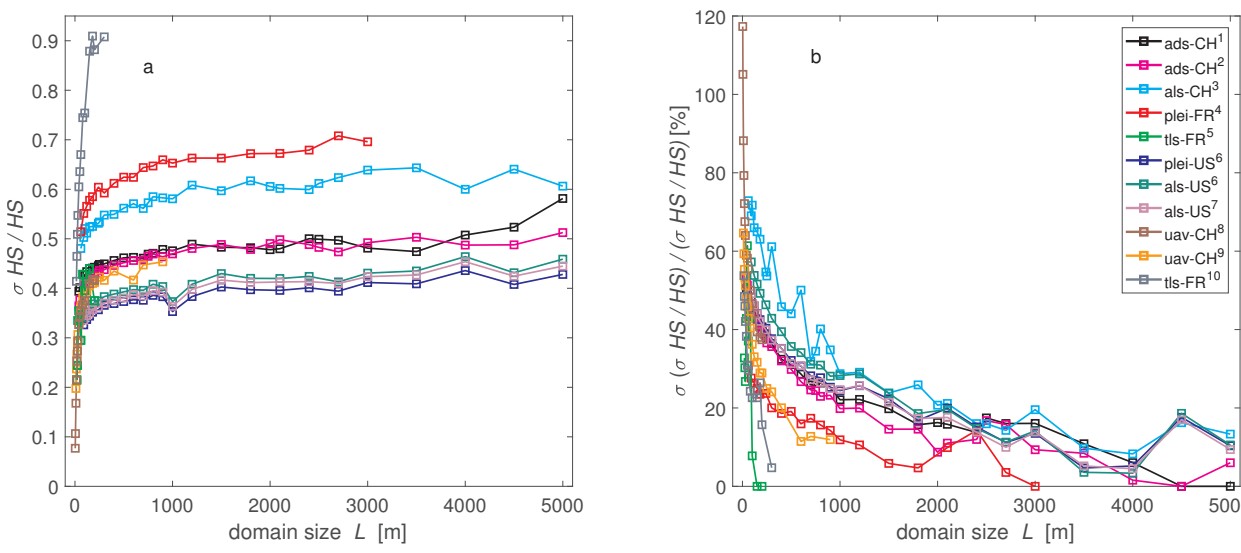

**Figure 6.** (a) Standard deviation of $HS$ as a function of domain size $L$ for each data set separately. (b) Percentage of standard deviation of (a) among each $L$. $L$ ranges from 3 m to 5 km. Colors represent the different geographic regions, measurement platform or acquisition dates (number) of the compiled data set as indicated in Section 2.1 to 2.4.

### 3.2 Autocovariances for scale breaks

The spatial autocovariance allows finding spatial scale breaks up to which snow depth values are highly correlated, i.e. up to which length scale the snow depth distribution is strongly dictated by local topographic interactions of the snow cover with wind, precipitation and radiation. Below this scale break process models should ideally explicitly resolve these interactions to

reliably describe the spatial snow depth distribution. Above this scale break we assume that dominant wind or precipitation patterns due to larger scale topography impacts dictate spatial snow depth distributions. At this scale range the normalized standard deviations of snow depth $\sigma_{HS}$ start levelling out (Figure 6a) as well as the normalized variability of $\sigma_{HS}$ among similar sized $L$ (Figure 6b).

We calculated spatial autocovariances for snow depth data sets with the "Fast Fourier Transform" (FFT), which allows computing spatial autocovariances up to large distances by keeping the fine grid cell resolutions. We used the R function fft() of the 'stats' package (see R Core Team, 2020).

### 3.3 Fractional snow-covered area parameterization (Helbig et al., 2015)

Helbig et al. (2015) derived a $fSCA$ parameterization by integrating a normal pdf assuming spatially homogeneous melt.

Subsequent fitting over a range of coefficients of variation $CV$ (standard deviation divided by its mean) between 0.06 and 1.00 resulted in a similar closed form fit for $fSCA$ as Essery and Pomeroy (2004) obtained by integrating a lognormal pdf:

$$fSCA = \tanh(1.3 \frac{HS}{\sigma_{HS}}) \,, \tag{1}$$



using current $HS$ and standard deviation of previous maximum snow depth or peak of winter. The standard deviation of snow depth at peak of winter was derived by relating peak of winter high-resolution spatial snow depth data from Switzerland and Spain to underlying summer terrain parameters (Helbig et al., 2015)

$$\sigma_{HS} = HS^a \mu^b \exp[-(\xi/L)^2] \tag{2}$$

with $a = 0.549$, $b = 0.309$ and $HS$ and terrain correlation length $\xi$ in meters. $\xi$ and $\mu$ are summer terrain parameters, where $\mu$ is related to the mean squared slope via $\mu = \left\{ \overline{[(\partial_\mathrm{x} z)^2 + (\partial_\mathrm{y} z)^2]}/2 \right\}^{1/2}$ using partial derivatives of subgrid terrain elevations $z$, i.e. from a DEM. The correlation length $\xi$ or typical width of topographic features in a domain size $L$ was derived via $\xi = \sqrt{2}\sigma_z/\mu$ with the standard deviation of elevations $\sigma_z$. The $L/\xi$ ratio indicates how many characteristic topographic features of length scale $\xi$ are included in each $L$. Similar to Helbig et al. (2015), we linearly detrended the summer DEM before deriving the terrain parameters to unveil the correct terrain characteristics associated with the shaping process of the snow depth distribution at the corresponding scale. Using Eq. (1), $fSCA$ can thus be derived with grid cell mean snow depth from a snow model and grid cell mean subgrid terrain parameters derived from a fine-scale summer DEM.

## 3.4 Deriving a new scale-independent fractional snow-covered area parameterization

Helbig et al. (2015) showed that $fSCA$ performances increased with spatial scale and yielded best performance for spatial scales larger than 1000 m. Since the $fSCA$ parameterization was empirically developed on snow depth data from two geographic regions, here we reevaluated the scaling variables for the spatial variability of snow depth $\sigma_{HS}$ as well as the functional form of the parameterization using the large compiled $HS$ data set of this study. Various scaling variables were previously employed to capture $\sigma_{HS}$ in mountainous terrain. Helbig et al. (2015) selected snow depth $HS$, the squared slope related parameter $\mu$ and the $L/\xi$ ratio (Eq. (2)), Skaugen and Melvold (2019) used $HS$ and standard deviation of the squared slope, others used $\sigma_z$ as terrain parameter (e.g. Roesch et al., 2001). Here, we were interested in finding dominant scaling variables that correlate consistently across scales with $\sigma_{HS}$. We therefore analyzed the pearson correlation coefficient $r$ between various candidate parameters and $\sigma_{HS}$ as a function of spatial scale, i.e. domain size $L$.

## 3.5 Performance measures

The performance in this article is evaluated by the following measures: the root mean square error (RMSE), normalized root mean square error (NRMSE, normalized by the range of measured data (max-min) or the mean of the measurements for $fSCA$), mean absolute error (MAE), the mean absolute percentage error (MAPE, absolute bias with measured minus parameterized and normalized with measurements), the mean percentage error (MPE, bias with measured minus parameterized and normalized with measurements) and the pearson correlation coefficient $r$ as a measure for correlation. We also evaluate the performances by deriving the two-sample Kolmogorov-Smirnov test (K-S test) statistic values $D$ (Yakir, 2013) for the pdfs and by computing the NRMSE for Quantile-Quantile plots (NRMSE$_\mathrm{quant}$, normalized by the range of measured quantiles (max-min)) for probabilities with values in $[0.1, 0.9]$.





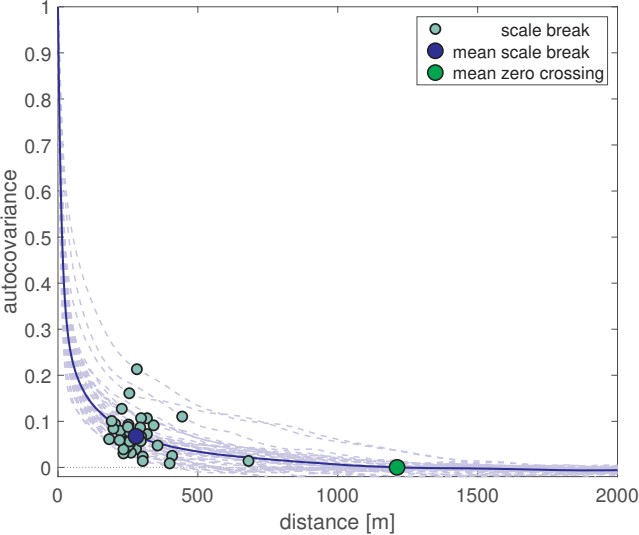

**Figure 7.** FFT derived autocovariances for spatial snow depth. Individual ranges, mean range and mean autocovariance zero crossing are shown.

## 4 Results

### 4.1 Spatial correlation range from snow depth data

We derived 40 autocovariances for domain sizes of 3 km with grid cell sizes $\Delta x$ of 2 or 3 m. By determining the corresponding inflection points for each domain size $L$ using R (R Core Team, 2020) we obtained scale breaks between 183 and 681 m with a mean of 284 m ($\pm \sigma$ 86 m) (Figure 7). The zero crossings for each $L$ were between 402 m and 1815 m with a mean of 1011 m ($\pm \sigma$ 402 m). For the mean autocovariance we obtained a scale break at about 279 m and a zero crossing at about 1212 m. Based on the observed scale breaks we selected a minimum length scale of 200 m for deriving a new scale-dependent $fSCA$ parameterization for all larger scales. In the following all results are therefore restricted to $L \geq 200$ m leaving a pool of 41'249 domain sizes $L$ with $L$ between 200 m and 5 km for the development of the parameterization.

### 4.2 Scaling variables for $\sigma_{HS}$

Correlation coefficients varied differently across spatial scales (Figure 8a). For all scales, we obtained the largest correlation coefficients for $HS$ ranging from 0.48 to 0.98 with a mean of 0.79. From correlations with the various subgrid terrain parameters, the largest correlations across all scales were reached for the squared slope related parameter $\mu$ ranging from 0.22 to 0.61 with a mean of 0.36. Similar consistent correlation coefficients across scales but slightly smaller for $L \leq 1800$ m resulted for the squared slope $sqS$ with an overall mean of 0.33 ($sqS$ is derived here from $2\mu^2$; cf. Section 3.3). The correlation coefficients for the standard deviation of $sqS$ ($\sigma_{sqS}$) and $\sigma_z$ were much less consistent across scales than for $\mu$ and $sqS$ and were overall lower. The mean correlation for $\sigma_{sqS}$ is 0.15, for $L/\xi$ 0.21 and for $\sigma_z$ 0.01. Though the mean correlation between $\sigma_{HS}$ and $L/\xi$



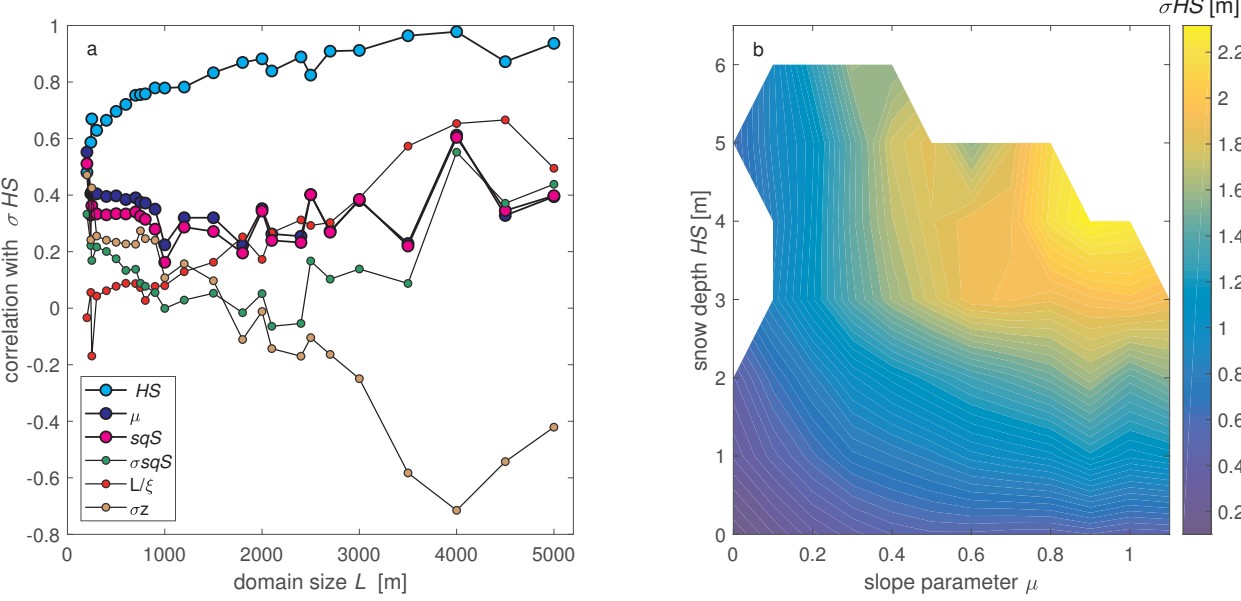

**Figure 8.** (a) Correlation coefficients between $\sigma_{HS}$ and various parameters as a function of domain size $L$. (b) Standard deviation of snow depth $\sigma_{HS}$ as a function of snow depth $HS$ and slope parameter $\mu$.

is rather low, correlation remains more consistent across scales up to about 2500 m and increases for larger scales considerably up to 0.67 (cf. Figure 8a).

We selected $HS$, $\mu$ and $L/\xi$ as main scaling parameters for $\sigma_{HS}$ across spatial scales from 200 m to 5 km (Figure 8b).

### 4.3  Scale-independent fractional snow-covered area parameterization

The correlation analysis across scales revealed the same dominant correlation parameters than in Helbig et al. (2015). We therefore kept the functional form for $\sigma_{HS}$ at peak of winter suggested by Helbig et al. (2015) using the three scaling variables $HS$, $\mu$ and $L/\xi$. The new $\sigma_{HS}$ parameterization at peak of winter thus has the same functional form than the one suggested by Helbig et al. (2015) which was presented in Eq. (2). However, the fit parameters $a$ and $b$ therein are replaced by new parameters $c$ and $d$ which we specify below. To derive the new parameters $c, d$ we fitted nonlinear regression models by robust

M-estimators using iterated reweighed least squares (see R (R Core Team, 2020) and its robustbase version 0.93-6 package (Maechler et al., 2020)).

Fit parameters $c, d$ were derived by randomly taking 500 sub-samples (80 %) from the snow depth data set. We derived $c, d$ scale-dependent for sample data starting with $L \geq 200$ m step wise up to $L \geq 5$ km (cf. individual colored lines in Figure 9). Scatter for the resulting $c, d$ increased with increasing $L$. Since the standard deviation among $c, d$ for $L \geq 200$ m was extremely

low with 0.001 for $c$ as well as for $d$ we first fitted constant parameters $c, d$ for the entire data pool and $L \geq 200$ m. We obtain constant fit parameters of $c = 0.6589 \ (\pm 0.0037)$ and $d = 0.5638 \ (\pm 0.0043)$ with the 90 % confidence intervals of the fit



**Figure 9.** Fit parameters for Eq. (2) as a function of domain sizes $L$ to scale variables (a) $HS$ and (b) $\mu$. Colored lines show the fit parameters derived by taking 500 random 80 % samples from the compiled snow depth data set. The dark blue dots depict the ensemble median. Previously obtained constant parameters of Helbig et al. (2015) (light blue dots) and newly fitted constant (red dots) as well as newly fitted scale-dependent (pink circles) parameters are shown.



parameters given in parentheses. These 'new' constant parameters $c, d$ are larger than the previously derived constants $a, b$ in Eq. (2) (cf. Figure 9). Given that the values of $c, d$ clearly increase with spatial scale $L$ (Figure 10) we introduced $L$ in $c, d$ to improve the application of Eq. (2) across scales. By fitting the ensemble median of the 500 random sub samples (dark blue

dots in Figure 9) we obtained scale-dependent parameters $c(L)$ and $d(L)$. We started at the scale length of 200 m, defined by the scale break which we derived before from spatial snow depth autocovariances. Fitting over samples larger than the corresponding $L$ instead of over samples at a specific $L$ should allow describing the combined larger scale topography-wind-precipitation impacts on the spatial snow depth distribution in mountainous terrain acting at scales larger than the observed scale break of about 200 m. Thus, Eq. (2) using the following scale-dependent parameters $c(L)$ and $d(L)$ assembles our new

$\sigma_{HS}$ parameterization for $L \geq 200$ m:

$$
\begin{aligned}
c(L) &= 0.5330\, L^{0.0389} \\
d(L) &= 0.3193\, L^{0.1034}
\end{aligned}
\tag{3}
$$

with the 90 % confidence intervals of $\pm 0.0097$, $\pm 0.0026$ and $\pm 0.0183$, $\pm 0.0079$ in the order of introduced constants in Eq. (3). The new $\sigma_{HS}$ parameterization using $c(L)$ and $d(L)$ (Eq. (2) with Eq. (3)) is applied in the previously derived $fSCA$ parameterization (Eq. (1)). To demonstrate the resulting differences when using scale-dependent versus scale-independent fit

parameters in parameterized $\sigma_{HS}$ (Eq. (2) we will also validate the performance using constant $c, d$ in the previously derived $fSCA$ parameterization as well as in the $\sigma_{HS}$ parameterization.

## 4.4 Evaluation

### 4.4.1 Evaluation for $\sigma_{HS}$ and $fSCA$ for all $L$

Parameterized $\sigma_{HS}$ and $fSCA$ perform well for all domain sizes, i.e. for $L \geq 200$ m of the entire data pool. Very similar

performance measures are obtained for the parameterizations using the newly derived constant fit parameters $c, d$ and the parameterizations using the scale-dependent parameters $c(L), d(L)$ (cf. Table 1 and I(a) and II(a)). We obtain a slightly better MPE for $\sigma_{HS}$ when using scale-dependent fit parameters (-4 % versus -5 %) however for $fSCA$ MPEs are the same (0.2 %). The same rather low NRMSEs result for $\sigma_{HS}$ (8 %) and for $fSCA$ (2 %).

### 4.4.2 Scale-dependent evaluation for $\sigma_{HS}$ and $fSCA$

While mean performance measures of the $\sigma_{HS}$ and $fSCA$ parameterization are almost uninfluenced to using constant or scale-dependent fit parameters (cf. Table 1 and I(a) and II(a)) we found diverging performances when analyzing performance measures as a function of scale (Figure 10). Across scales, improved or similar performances were achieved when using scale-dependent fit parameters in parameterized $\sigma_{HS}$ especially for larger scales. Maximum performance improvements of 4 % occurred for $L$ of 2500 m, respectively for $fSCA$ of 0.7 % when using scale-dependent fit parameters. Thus, introducing

scale-dependent fit parameters enhanced the $\sigma_{HS}$ parameterization for application across scales.





**Table 1.** Performance measures for all $L$ between measurement and parameterization of (I) standard deviation of snow depth $\sigma_{HS}$ with (a) Eq. (2) and constant or $L$ dependent fit parameters $c, d$ (Eq. (3)) and (b) $\sigma_{HS}$ as in Helbig et al. (2015); Skaugen and Melvold (2019) and of (II) $fSCA$ with (a) Eq. (1) and (Ia) and (b) $fSCA$ as in Helbig et al. (2015) and Skaugen and Melvold (2019) using Eq. (1).

| | NRMSE | RMSE | MPE | MAPE | MAE | $r$ | K-S | NRMSE$_{quant}$ |
|---|---|---|---|---|---|---|---|---|
| | [%] | [cm] | [%] | [%] | [cm] | | | [%] |
| **I $\sigma_{HS}$** | | | | | | | | |
| (a) Eq. (2) with | | | | | | | | |
| constant $c, d$ parameter | 7.9 | 26.6 | -5.3 | 22.6 | 19.7 | 0.83 | 0.05 | 5.3 |
| $c(L), d(L)$ (Eq. (3)) | 7.9 | 26.7 | -4.1 | 22.4 | 19.6 | 0.83 | 0.05 | 5.5 |
| (b) previous parameterizations from | | | | | | | | |
| Helbig et al. (2015) | 9.3 | 31.1 | -29.5 | 36.7 | 25.3 | 0.82 | 0.22 | 14.6 |
| Skaugen and Melvold (2019) | 20.4 | 68.5 | -77.9 | 82.8 | 57.9 | 0.68 | 0.48 | 37.6 |
| | NRMSE | RMSE | MPE | MAPE | MAE | $r$ | K-S | NRMSE$_{quant}$ |
| | [%] | | [%] | [%] | | | | [%] |
| **II $fSCA$** | | | | | | | | |
| (a) Eq. (1) with | | | | | | | | |
| Eq. (2) and constant $c, d$ parameter | 2.4 | 0.02 | 0.22 | 1.11 | 0.01 | 0.64 | 0.37 | 0.5 |
| Eq. (2) and $c(L), d(L)$ (Eq. (3)) | 2.4 | 0.02 | 0.16 | 1.09 | 0.01 | 0.63 | 0.37 | 0.4 |
| (b) previous parameterizations from | | | | | | | | |
| Helbig et al. (2015) | 3.2 | 0.03 | 1.45 | 1.8 | 0.02 | 0.74 | 0.47 | 1.6 |
| Skaugen and Melvold (2019) using Eq. (1) | 6.2 | 0.06 | 3.87 | 4.8 | 0.05 | -0.04 | 0.75 | 4.4 |

### 4.4.3 Scale- and region-dependent evaluation for $\sigma_{HS}$ and $fSCA$

A large data set from various geographic regions allows us to develop a more reliable empirical parameterization than being limited to the characteristics by a few data sets. Here, we not only compiled data sets from various geographic regions but the data sets were also acquired by different measurement platforms coming with a range in inaccuracies between below 10 cm to 305 80 cm. As a consequence larger scatter in performances appear when performance measures are depicted not only as a function of spatial scale but also region wise, including platform wise. While most of the MPEs are still between -20 % and 10 % some regions strike out because they have much larger MPEs when binned scale as well as region wise (Figure 11). For instance a MPE of up to 60 % for $\sigma_{HS}$ was obtained for TLS data from the southeastern French Alps and overall larger MPEs, though consistent across scales, for the Pléiades data from the northeastern French Pyreenes. MPEs for $fSCA$ on the other hand do 310 not show a similar large spread among the regions and are low between -1 % to 2 % (Figure 11b).





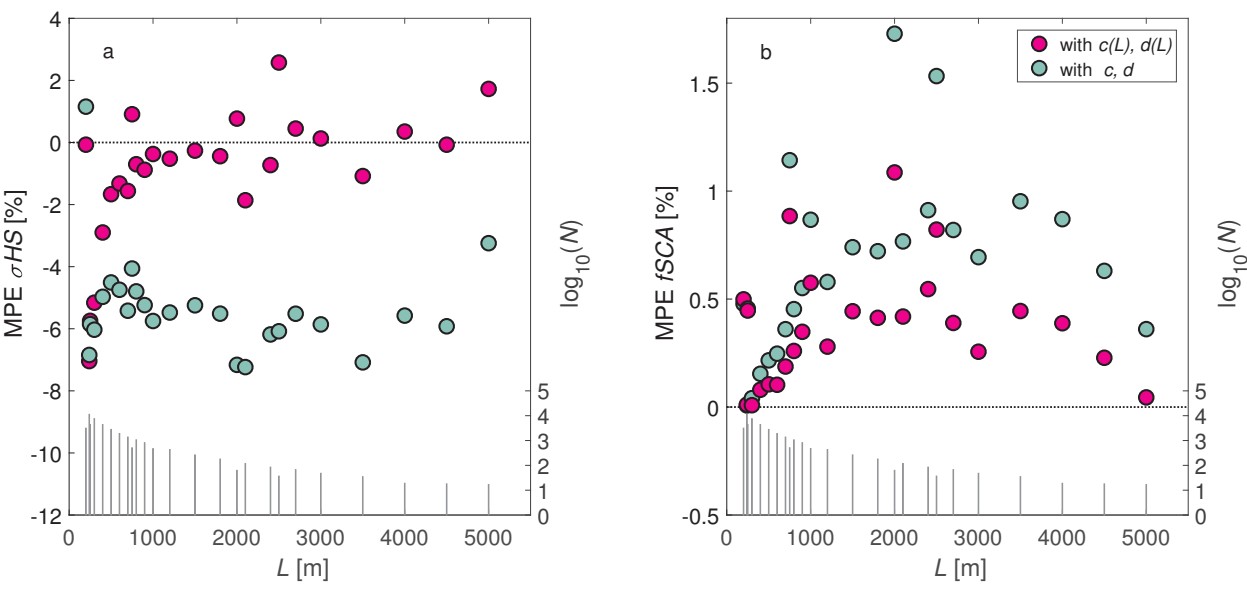

**Figure 10.** Mean percentage error (MPE) as a function of $L$ for (a) $\sigma_{HS}$ and (b) $fSCA$. MPEs are shown for the $\sigma_{HS}$ and $fSCA$ parameterizations using Eq. (1) to (3) with scale-dependent $c(L), d(L)$ as well as for constant $c, d$. The second y-axis shows the number of valid domains per $L$ on a logarithmic scale.

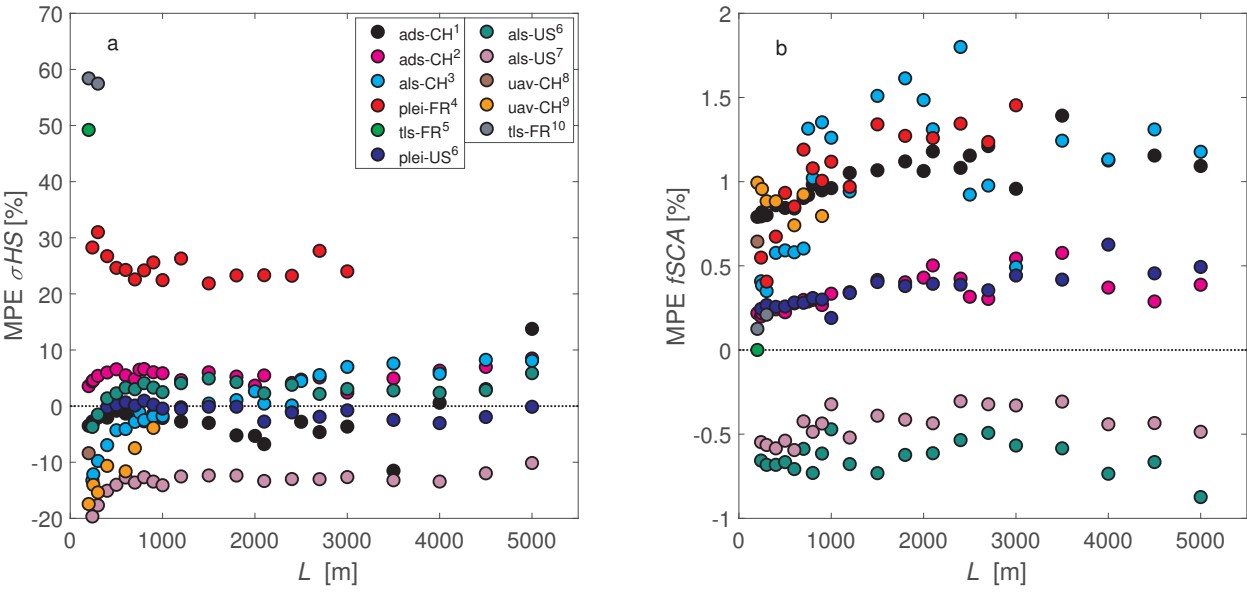

**Figure 11.** Mean percentage error (MPE) as a function of $L$ for the compiled data set for (a) $\sigma_{HS}$ and (b) $fSCA$ using Eq. (1) to (3) with scale-dependent $c(L), d(L)$. Colors represent the different geographic regions, measurement platform or acquisition dates (number) of the compiled data set as indicated in Section 2.1 to 2.4.



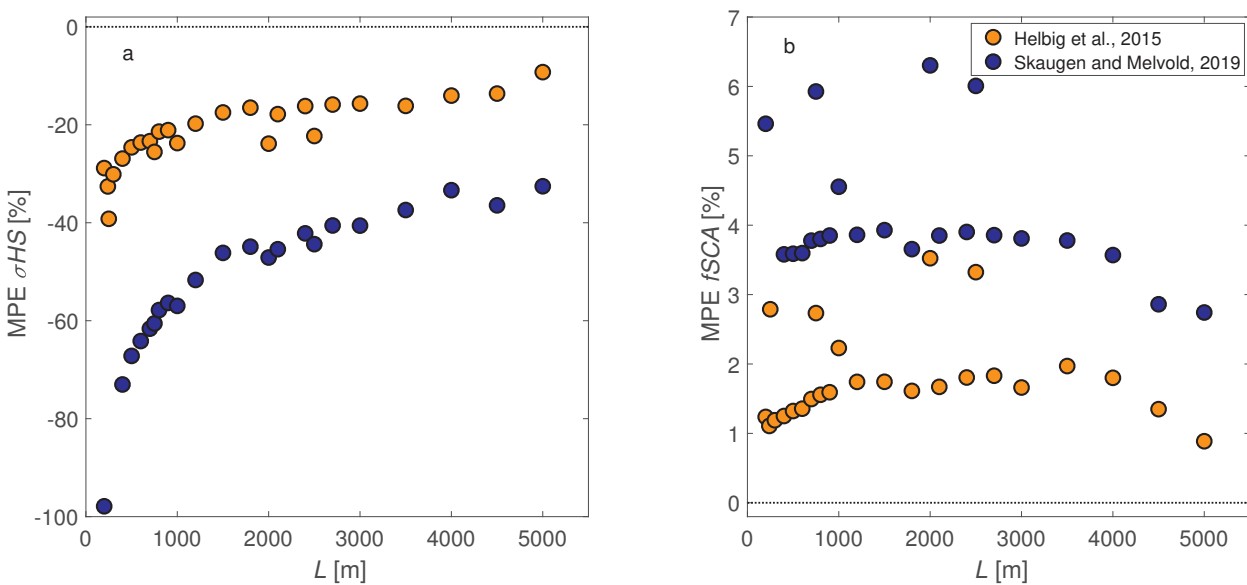

**Figure 12.** Mean percentage error (MPE) as a function of $L$ for the compiled data set for (a) $\sigma_{HS}$ and (b) $fSCA$. MPEs are shown for the $\sigma_{HS}$ and $fSCA$ parameterizations of Helbig et al. (2015) as well as for the $\sigma_{HS}$ parameterization of Skaugen and Melvold (2019) and Skaugen and Melvolds $\sigma_{HS}$ parameterization applied in the $fSCA$ parameterization of Helbig et al. (2015) (Eq. (1)).

### 4.4.4 Evaluation of previous closed form parameterizations

To increase our understanding of the performances achieved with the new parameterizations, we also tested two previously derived empirical parameterizations. Specifically we investigated how parameterized $\sigma_{HS}$ using Eq. (2) (Helbig et al., 2015) and using the recently published formulation of Skaugen and Melvold (2019) compare to observed $\sigma_{HS}$ of our compiled data

set (Figure 12a). We further tested both $\sigma_{HS}$ parameterizations in the $fSCA$ parameterization (Eq. (1); Figure 12b). The parameterization of Helbig et al. (2015) works well. The performance measures for all $L$ are only slightly worse compared to the new parameterizations using both constant as well as scale-dependent fit parameters (Table 1). However, compared to the performance measures for the parameterization of Skaugen and Melvold (2019) the performances of Helbig et al. (2015) are clearly improved. Though MPEs of both previous $\sigma_{HS}$ parameterizations are scale-dependent, the MPEs of Skaugen

and Melvold (2019) reveal a larger scale-dependence of the performances compared to Helbig et al. (2015) (Figure 12a). In particular, individual MPEs vary a lot from MPEs for all $L$ given in Table 1.



## 5 Discussion

### 5.1 Spatial correlation range

While multi-scale behaviour for spatial snow depth data has been found in various studies, observed scale breaks depend on the extent and horizontal resolution of the investigated snow depth data sets. A first scale break of spatial snow depth data in treeless, alpine terrain has been observed between 10 to 20 m (e.g. Deems et al., 2006; Trujillo et al., 2007; Schweizer et al., 2008; Schirmer and Lehning, 2011; Mendoza et al., 2020) and a second scale break has been observed at around 60 m (Trujillo et al., 2009). By computing spatial autocovariances starting with domain sizes $L$ of 200 m in 0.1 m to 1 m resolution up to 3 km in 2 to 3 m resolution we also detected the two previously found scale breaks [not shown]. However, by additionally covering larger spatial extents than previously have been investigated, we also detected a third scale break with a mean at about 280 m (Figure 7). A similar scale break at around 200 m was recently found by analyzing performance decreases of distributed snow modelling in various grid cell sizes together with semivariogram analysis of subgrid summer terrain slope angles in the same catchment in the High Atlas (Baba et al., 2019). While for other application studies, such as in avalanche forecasting the smaller scale breaks are decisive for explicitly describing the relevant snow cover processes, here we are more interested in the largest detected scale break. At these scale lengths the longer range processes of precipitation, wind and radiation interactions with topography most dominantly influence the spatial snow distribution in mountainous terrain, which we assume can be parameterized with sufficient accuracy at this length scale by a scale-independent parameterization.

### 5.2 Scaling parameter

We not only investigated dominant correlations between the spatial snow depth distribution and terrain parameters but we also analyzed these correlations as a function of spatial scale. For some commonly applied scaling parameters this revealed large variations of correlations across scales such as for $\sigma_z$ (Figure 8a). Similar to our results, Skaugen and Melvold (2019) also obtained large correlations between $\sigma_{HS}$ and mean squared slope $sqS$ for spatial snow depth data sets acquired at peak of winter in Norway though this was only analyzed for spatial scales of 0.5 km x 1 km. Nevertheless, this confirms our findings since mean squared slope is related to the slope related parameter $\mu$ used here by $sqS = 2\mu^2$. However, Skaugen and Melvold (2019) obtained slightly improved correlation for the standard deviation of squared slope and therefore selected this parameter to stratify the topography for parameterizing $\sigma_{HS}$. Across spatial scales as well as for all $L$ we obtained lower correlations between the standard deviation of squared slope and $\sigma_{HS}$ though we observed cross-correlations between mean and standard deviation of squared slope of 0.71 indicating that both parameters correlate well with $\sigma_{HS}$.

### 5.3 Scale-independent $fSCA$ parameterization

The closed form fractional snow-covered area parameterization $fSCA$ given in Eq. (1) got enhanced by recalibration and introducing scale-dependent fit parameters (Eq. (3)) to make the performance consistent across spatial scales.





We developed the parameterization on a large snow depth data set. Large variability in the snow depth data set was gained by compiling 11 individual data sets from varying geographic regions as well as various measurement platforms. While the latter might explain remaining performance differences discussed below, the first led to large variability in summer terrain
characteristics and snow climates and consequently spatial snow depth distributions (cf. Figure 2). Though our presented parameterization for $\sigma_{HS}$ was empirically derived it is reassuring that for a new empirical derivation on a much larger and more diverse snow depth data set the same underlying functional form could be used. Furthermore, larger (about 17 % respectively 45 % larger) but overall consistent constant fit parameters were obtained compared to the derivation on the limited number of two data sets in two geographic regions by Helbig et al. (2015) (cf. $a, b$ in Eq. (2) and $c, d$ presented in Section 4.3 or Figure 9).
In addition to deriving constant fit parameters across spatial scales we took 500 random sub samples from the compiled snow depth data set to which we fitted scale-dependent constants (Figure 9). Scale-dependent constants considerably increased with increasing scale from $L$=200 m to $L$=5 km by at most 12 % respectively 38 % (Figure 9). This demonstrates that accounting for scale-dependent constants in the $fSCA$ parameterization (Eq. (1) with Eq. (2) and Eq. (3)) had to be performed. While we did not split our data set in development and validation subset, fitting over the ensemble median of the 500 sub samples to
derive $c(L), d(L)$ ensures confidence in the resulting fit parameters.

An increase in scatter among all $c(L)$ and $d(L)$ with increasing domain scale $L$ (Figure 9) can be most likely explained by a concurrent decrease in available valid data in larger $L$. Though we required at least 70 % valid data per $L$ when aggregating $HS$ data in domain sizes $L$, the maximum threshold of 70 % was more often required for the larger $L$ than for smaller $L$.

## 5.4    Evaluation

### 5.4.1    Evaluation for $\sigma_{HS}$ and $fSCA$


Upon deriving performance measures on parameterized and observed $\sigma_{HS}$ and $fSCA$ for all $L$ (i.e. the pooled performance) we obtained very similar performances when using newly derived constant or scale-dependent fit parameters, i.e. $c, d$ or $c(L), d(L)$ (Table 1). Despite considerable differences up to 12 % for $c$ and up to 38 % for $d$ between constant and scale-dependent fit parameters (Figure 9) pooled performances for all $L$ for $\sigma_{HS}$ and $fSCA$ were similar (Table 1). An explanation
for this is that the number of available domains is strongly decreasing with increasing $L$. For $L \geq 3000$ m we have only about 0.33 % (137 in total) valid domains available compared to the total of 41'249 for $L <$ 3000 m (Figure 4). This emphasizes the need for a scale-dependent evaluation.

### 5.4.2    Scale-dependent evaluation for $\sigma_{HS}$ and $fSCA$

While the largest improvement in MPE for all $L$ seem to origin from recalibration using the new compiled data set with a
reduction in MPE from -30 % to -5 % compared to a reduction from -5 % to -4 % when introducing scale-dependent fit parameters (Table 1), MPEs as a function of scale clearly demonstrated the improved behaviour when using scale-dependent $c(L), d(L)$ instead of constant fit parameters $c, d$ in the $\sigma_{HS}$ and $fSCA$ parameterization (Figure 10). Given that constant $c, d$ were fitted over the entire data set as have been $c(L), d(L)$, any performance improvement using $c(L), d(L)$ instead of constant



$c, d$ for parameterized $\sigma_{HS}$ and $fSCA$ origins in introducing scale-dependent parameters. For the parameterizations using the

constant fit parameters $c, d$ errors varied slightly more across scales than when using the scale-dependent $c(L), d(L)$ version. Individual scale-dependent errors were in part larger than the MPEs for all $L$ given in Table 1. Unequal numbers of valid domains per $L$ most likely also contributed to this.

### 5.4.3   Scale- and region-dependent evaluation for $\sigma_{HS}$ and $fSCA$

Studying region-wise performances reveals the spread in errors we can expect when the new parameterizations are applied
on an individual independent data set (Figure 11). We obtain much larger positive MPEs for $\sigma_{HS}$ at lower spatial scales of $L =$200 m and $L =$300 m for the two TLS data sets in the southeastern French Alps and overall larger MPEs between 20 to 30 %, though consistent across scales, for the Pléiades data above the Bassiès basin in the northeastern French Pyrenees. It is unclear if these larger MPEs origin in uncertainties of the data acquisition, i.e. are platform specific, or if they are linked to spatial snow depth distributions which could not be captured by the proposed new parameterizations. RMSEs for the various
remote sensing platforms and data sets used here (Section 2) descend from 80 cm for Pléiades data from the Sierra Nevada, to 33 cm for the ADS, to 16 cm for UAS, to 13 cm for ALS data from Switzerland, to 8 cm for ALS data from the Sierra Nevada and to 4 to 10 cm for TLS data in general. Given the rather low errors typically obtained for TLS data compared to the other remote sensing platforms, the reason for the large deviations of the TLS data sets might not origin in inaccuracies of the data acquisition. On the contrary, the observed bias in the Pléiades data from the northeastern French Pyrenees might
be attributable to platform connected rather large inaccuracies with NMADs of 45 cm to 78 cm (Marti et al., 2016). However, Pléiades data from the Sierra Nevada comes with a similar large NMAD of 69 cm but $\sigma_{HS}$ can be parameterized very well with MPEs lower $\pm 3$ % across spatial scales (Figure 11a). Observed $\sigma_{HS}$ from the TLS as well as from the Pléiades data in France was considerably larger than parameterized $\sigma_{HS}$ but mean slope angles alone can also not explain this behaviour (between 6 and 23° for the TLS data and between 13 and 50° for the Pléiades data).

### 5.4.4   Evaluation of previous closed form parameterizations

Though we developed a new peak of winter $\sigma_{HS}$ parameterization (Eq. (3)), empirically derived parameterizations can only describe the variability inherent in the data set used to derive the parameterization. In addition to the region-wise evaluation, analyzing performances of previous empirically derived parameterizations may therefore allow estimating expected performance sensibility to independent data sets. While both tested parameterizations of $\sigma_{HS}$ (Helbig et al. (2015); Skaugen and
Melvold (2019)) showed worse performances than the new parameterizations and less consistency as a function of scale, the model performances of Helbig et al. (2015) were only slightly worse than the new parameterizations (Table 1). Since only one out of the 11 data sets used in this study was previously used to develop the parameterization of Helbig et al. (2015), an overall similar performance of Helbig et al. (2015) (Figure 12) with the large compiled data set of this study clearly confirms the underlying functional form of $\sigma_{HS}$ suggested by Helbig et al. (2015) which was reapplied here.



## 6 Conclusions

We presented an empirical peak of winter parameterization for the standard deviation of snow depth $\sigma_{HS}$ for treeless, mountainous terrain describing the spatial snow depth distribution in a grid cell for various model applications. The scaling variables of the new parameterization of $\sigma_{HS}$ and $fSCA$ are the same than in Helbig et al. (2015) which are spatial mean snow depth, a squared slope related parameter and a terrain correlation length. All subgrid terrain parameters can be easily derived from fine-scale summer DEMs for each coarse grid cell.

By introducing spatial scale dependencies in the variables of the formulation for $\sigma_{HS}$ of Helbig et al. (2015), $\sigma_{HS}$ can be consistently parameterized across spatial scales starting at scales $\geq$200 m. The spatial snow depth variability or $\sigma_{HS}$ is the important variable to parameterize the fractional-snow covered area $fSCA$ (Helbig et al., 2015). Performance improvements across spatial scales of the $\sigma_{HS}$ parameterization therefore directly enhanced the $fSCA$ parameterization. Between length scales of 200 m and 5 km mean percentage errors (MPE) were between -7 % and 3 % for $\sigma_{HS}$ and between 0 % and 1 % for $fSCA$.

The subgrid parameterization of $\sigma_{HS}$ was developed on a large pool of 11 spatial snow depth data sets from 7 different geographic regions in high spatial resolutions between 0.1 m to 3 m and with spatial coverage between 0.14 to 280 km$^2$. An evaluation of two previously presented empirical $\sigma_{HS}$ parameterizations confirmed the functional form of the parameterization of Helbig et al. (2015) as well as the need to enhance its performance across scales. By analyzing data from the large pool of 11 spatial snow depth data sets, we were able to recalibrate the subgrid parameterization of $\sigma_{HS}$ and achieved improved performances using new constant fit parameters. Additionally introducing a scale-dependency in the dominant scaling variables further improved the performance across spatial scales. Mean MPEs of $\sigma_{HS}$ over all scales (i.e. pooled performance) reduced from -30 % using Helbig et al. (2015) to -5 % after recalibration to -4 % after introducing scale-dependent fit parameters (Table 1). Individual scale-dependent improvements in MPEs reached up to 4 % when using newly derived scale-dependent fit parameters compared to newly derived constant fit parameters for $\sigma_{HS}$ on the large data pool. This shows the improvement thanks to introducing scale-dependent parameters (Figure 10). Towards estimating the possible spread in performances when applying empirically derived $\sigma_{HS}$ and $fSCA$ for independent geographic regions we validated the parameterizations region- and scale-specific. While this clearly increased MPEs for three data sets, the majority of the region- and scale-dependent MPEs were between $\pm$ 10 % for $\sigma_{HS}$ and between -1 % and 1.5 % for $fSCA$ indicating that the parameterizations perform similar well in most geographical regions.

A peak of winter parameterization of $\sigma_{HS}$ describes the maximum spatial snow depth variability during a winter season which is of interest for various model applications. A peak of winter parameterization can however not alone be used to describe the seasonal $fSCA$ evolution because a reliable model application of any $fSCA$ parameterization requires an implementation accounting for alternating snow accumulation and melt events during the season, i.e. to describe the $SCD$. Especially at lower elevations the separation of the $SCD$ in only one accumulation period followed by a melting period is no longer valid (Egli and Jonas, 2009). A description of an algorithm for a seasonal $fSCA$ model implementation which uses the new scale-independent peak of winter $fSCA$ parameterization presented here is currently in preparation. Extending the empirical peak of



winter $fSCA$ parameterization to a broader range of scales and snow climates was thus a meaningful step towards accounting
450  for spatiotemporal variability in snow depth for multiple snow model applications.

*Data availability.* All data used in this study is described in the data section. The data can be downloaded from the referenced repositories
or data availability is described in the referenced publications.

*Author contributions.* YB, LE, CDB, SG, MD, JR, JSD, TJ: data acquisition/processing, experiment/platform design and setup; NH: development of parameterization; All authors contributed to the paper with discussions and ideas. NH wrote the paper with contributions from all
455  co-authors.

*Competing interests.* The authors declare that they have no conflict of interest.

*Acknowledgements.* N. Helbig was funded by a grant of the Swiss National Science Foundation (SNF) (Grant N° IZSEZ_186887), as well
as partly funded by the Federal Office of the Environment FOEN. Pléiade imagery was acquired through DINAMIS (Dispositif Institutionnel
National d'Approvisionnement Mutualisé en Imagerie Satellitaire). S. Gascoin acknowledges support from CNES Tosca and Programme
460  National de Télédétection Spatiale (PNTS, http://www.insu.cnrs.fr/pnts), grant n°PNTS-2018-4. Lautaret data were acquired thanks to J.
Revuelto AXA grant and to ANR JCJC EBONI (grant number ANR-16-CE01-006). CNRM/CEN is part of Labex OSUG@2020 (investissement d'avenir – ANR10 LABX56).



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
