# Peer review of "Fractional snow-covered area: Scale-independent peak of winter parameterization"

_The Cryosphere, 2020_

## Referee Comment (RC1) · Anonymous Referee #1 · 7 Oct 2020

This paper provides updated calibration and validation of a previously proposed model for fractional snow cover in mountain regions as a function of snow depth, its standard deviation, and two alpine terrain parameters (mean slope and horizontal feature correlation length). Whereas previous work calibrated the model based on snow depth and terrain information from only a couple of geographic locations, the current study pools information from a much broader range of sites using multiple observation techniques.

General Comments

My largest concern is what portion of the data set was used to calibrate the model (Section 4.3) vs evaluate the model in Section 4.4.1. The description of the methods in Section 4.3 wasn't clear (see further points made below) and seems to suggest that 80% of the data was used to calibrate the model and then the results in Table 1 and

Figure 10 were obtained using 100% of the data (or was it the remaining 20%?). Please clarify these points. If this is indeed the case, I don't think it's appropriate to refer to the results in Section 5.4.1 as an "evaluation."

Since the data set wasn't split for calibration and validation (line 364), I think the evaluation by region is particularly important and it will provide a better sense of how differently new data is likely to perform compared to pooled performance of the calibrated model. You state this explicitly on line 389, but it could be emphasized more. It also indicates that climate models would benefit from observations across as broad a selection of alpine regions as possible. While this study uses data from across a greater number of regions than your previous work, it may still not sample enough regions to be applicable to global mountain snow. Is it possible to provide a sense to the reader (perhaps in the discussion) of how well the distributions shown in Figure 5c represent values from global snow-covered mountain ranges? What about how much variability is there in $\xi$ across your pooled data vs globally? Would interannual variability in snow in a particular mountain region affect the values calculated for $\sigma HS$? I'm guessing it wouldn't if the snow and terrain is deep but perhaps in particularly low water years it might.

While you cite Helbig et al., 2015 and Essery and Pomeroy, 2004 in the introduction there is a lot of context from those two papers which is essential to understand this current study, and I think this manuscript would benefit from including more thorough explanations and context from them. I've tried to specify several examples in the comments below which I think would help but there may be others.

Specific comments

L100-106: I think the terminology "peak of winter fSCA" parametrization causes some confusion here. I recognize that sigma_HS is being calibrated by mid-winter (March) data, but from your previous work (Helbig 2015) you are expecting that the parametrization will apply during accumulation and melt as well. Furthermore the fSCE you describe in eq 2 is calculated from Helbig et al. (2015) assuming melt events resulting in a SCD curve. This paper never mentions the accumulation season or melt until the very end at line 445. Unless you have changed your opinion on whether or not the formulation used here and in Helbig et al., 2015 can be extrapolated outside of the peak season, I think you should mention this at some point in the introduction. If you are truly concerned that it can't be extrapolated outside of the peak-snow season I think you need to justify its potential use.

A visual representation of L, $\xi$, dx,dy would be helpful. E.g. representational lines on Figure 3, another panel in Figure 3, or at least explicitly refer the reader to previous work (e.g. Fig 2 of Helbig et al., 2009).

Equation 1: I think it would be helpful to state that equation 1 has been shown to reasonably parametrize fSCA for both nonmoutainous and mountainous regions, while the relationship in Eq 2 is derived using only mountain data (at approx. seasonal peak, if you'd like). And/or state this distinction in the introduction (e.g. at line72, "While the standard deviation of snow depth introduced by Essery and Pomeroy did not depend on subgrid terrain characteristics, the formulation shown in Equation 2 was introduced by Helbig et al. (2015) in order to better model Equation 1 in mountainous terrain."

You refer the reader to Helbig et al 2015 at the start of Section 3, but it's not clear if this is to describe the domain sampling procedure, or even if the same method used in the 2015 paper is used in this manuscript. The 2015 reference specifies 12 domain sizes between 50 and 3000m were randomly sampled. In this manuscript there are 20 bins shown on Fig 4. Please provide additional information on how each data set/scene is decomposed into domain sizes.

L207: The symbol HS is being used to represent both the domain-average snow depth and the high-resolution observed snow depths at fine scale resolution (e.g. figures 5a, 6a, lines 73-100). I suggest you distinguish these uses.

L240: Do you sample the autocovariance in each domain 40 times? Why do you single

out L=3km and then say you find inflection points for each domain size L?

L253: This is the first time you use sqS, and sigma_sqS. Again for context it would be good to mention that you are repeating previous analysis that established $\mu$ and $\xi/L$ as the most important correlates, and you are examining these two variables to compare to results from Skaugen and Melvold, which you do in the Discussion section.

L267-280: While I understand the results shown in Figure 9, I couldn't understand your description of the methods used to produce them. I suggest removing/reordering the first 4 sentences from this paragraph. The discussion of domain size dependent fitting only confuses things when you then discuss the fit to the entire pooled data set. I suggest beginning the paragraph with "Fit parameters were first calibrated for the entire data pool yielding c = 0.6589 ($\pm$0.0037) and d = 0.5638 ($\pm$0.0043) with the 90 % confidence interval. . . . . . . . . larger than the previously derived constants a, b in Eq. (2) (cf. Figure 9). For each step-wise domain size between 200 m to 5 km scale-dependent parameter values are also fit from the data (cf. individual colored lines in Figure 9)." At this point please provide a more complete description of the sub-sampling used to derive c, d for each step-wise domain size. What does 80% mean? Are the parameter values fitted from all the data within a randomly chosen domain of the appropriate size and this process is repeated 500 times? For domain sizes above 1km there are <500 domains total so are the same values just replotted? After this description, you can continue on with the discussion of how the parameters increase with L and the subsequent fitting of c(L) and d(L).

Fig 9: Please use a different description on the legend in place of 'f(L) – Eq. (3)' which can read as 'f(L) minus Equation (3).'

Section 4.4.4: Does the different choice of domain aspect ratio (square vs rectangular) affect the differences described in this section?

L335-337: Please rephrase for clarify: "at these scale lengths." I think you are saying something like "Above scale-lengths of $\sim$200m all three effects (precip/wind/radiationinteractions) come into play, while we think there are different physical effects which establish the breaks at 20 and 60m," but please confirm. Also consider rephrasing "scale-independent parameterization", since the parametrization incorporates scale information from the sub-domain terrain parameters as well as in the constants ( $c(L)$, $d(L)$ ). Perhaps something like "The results presented here indicate that the model described by (eqs. 1 and 2) is a reasonable fSCA parametrization in mountainous terrain for spatial scales between 200m to 5km." Given that you are aiming to have this used as a fSCA parametrization in climate models which can still use grid scales as high as ∼50-100km please comment on the extrapolation of your results substantially beyond 5km.

L343: Do you mean "for spatial scales between 0.5km and 1km"?

L357: "Furthermore, larger (about 17% and 45% , respectively) but overall consistent constant fit parameters were obtained compared to those from Helbig et al. (2015) based on a more limited number of data sets and just two geographic regions (cf. a, b. . ."

L411-413: I'd suggest that the appropriate standard for how different parametrizations perform is the range of MPE seen among different regions, not the difference between your previous calibration and the current one.

Technical Corrections

L185: 3m to 5km

Discussion, several places: "origin" as a verb -> "originate"

L379: I'd suggest splitting this sentence in two.

L400: rephrase

L395: "decrease from 80cm. . ."

L409: "sensitivity"?

---

## Referee Comment (RC2) · Anonymous Referee #2 · 29 Oct 2020

General Comments

The research presented is good progressive development of the lead author's previous research parameterizing fractional snow cover area. Improved empirical parameterization of sigma snow depth is presented in this research.

Specific Comments

The 11 diverse spatial, high-resolution snow depth data sets were pooled to develop an empirical parameterization for sigma snow depth. There is some discussion of how snow depth data from the sensors/platforms affect results in different sections of the paper. Can a summary of which sensor/platform provides the "best" snow depth data set resulting in a better parametrization for sigma snow depth?

[Figure]

Done with snow depth data sets at annual maximum snow cover, how might parameterization of sigma snow depth with data sets collected at mid-season or late season of snow cover affect the results? Is there a preferred time in relation to seasonal snow cover to collect a high-resolution snow depth dataset? Is it possible to use multiple snow cover data sets collected at a site at different times during the season to parameterize sigma snow depth?

Technical Corrections

Line 108: Suggest delete "large quantity", it is an unnecessary qualitative description of data used.

Line 137: The words "than for the" cause some confusion. Was the ALS data processed similar to the ASO campaigns or different from those campaigns?

Line 173: "lower zero" should be lower than 0, or snow depth $\leq$. "above" could be changed to $>$. And units should be given with "threshold of zero"

Line 228: "pearson" should be capitalized, prop noun. Applies throughout the paper.

Line 307: The expression "strike out" would be better stated as standout.

Line 398: Please clarify "not origin". Possibly originate is a word that could clarify source.

---

## Author Comment (AC1) · 19 Nov 2020

We thank the reviewer # 1 for the review and the constructive comments! All reviewer comments (in italics) are addressed below.

**General comments**

My largest concern is what portion of the data set was used to calibrate the model (Section 4.3) vs evaluate the model in Section 4.4.1. The description of the methods in Section 4.3 wasn't clear (see further points made below) and seems to suggest that 80% of the data was used to calibrate the model and then the results in Table 1 and Figure 10 were obtained using 100% of the data (or was it the remaining 20%?). Please clarify these points. If this is indeed the case, I don't think it's appropriate to refer to the results in Section 5.4.1 as an "evaluation."

We first splitted the data set in 25 sub pools for all available domain sizes L by using  $L \ge 200$  m,  $L \ge 240$  m etc. We then took 500 random samples from each of the 25 sub data pools but each sample comprised only 80 % of the sub pools data. Fitting each sample separately revealed the variance among the samples increasing with L (colored lines in Figure 9). Instead of fitting over 100 % of the data with  $L \ge 200$  m, we fitted over the ensemble median of all scale-dependent parameters derived from each of the random sample. This procedure should ensure sufficient robustness of the derived scale-dependent parameters c(L), d(L) (Eq. 3).

It is however true, that we did not compile an additional independent data pool for an independent evaluation and that we also did not split our compiled data set in development and validation subset. Given that we compiled and preprocessed a vast snow depth data pool covering large variability in snow climates and topographic characteristics, we were however able to perform a scale- as well as geographic region-dependent evaluation to reveal the spread in errors we can expect when applying the parameterizations on an individual independent data set (Figure 11).

We have rewritten Section 4.3 also considering the reviewers specific comment and suggestions below and further adapted Section 5.3.

Since the data set wasn't split for calibration and validation (line 364), I think the evaluation by region is particularly important and it will provide a better sense of how differently new data is likely to perform compared to pooled performance of the calibrated model. You state this explicitly on line 389, but it could be emphasized more. It also indicates that climate models would benefit from observations across as broad a selection of alpine regions as possible.

We agree that the scale- as well as region-dependent evaluation allowed the assessment of performances we can expect when applying the parameterizations on independent data sets. We now additionally mention this in the abstract. We also extended the available discussion on that in line 364. We did not add it to the conclusions, since it was already mentioned in the conclusions of the manuscript (line 437-441 of first submission).

While this study uses data from across a greater number of regions than your previous work, it may still not sample enough regions to be applicable to global mountain snow. Is it possible to provide a sense to the reader (perhaps in the discussion) of how well the distributions shown in Figure 5c represent values from global snow-covered mountain ranges? What about how much variability is there in  $\xi$  across your pooled data vs globally? Would interannual variability in snow in a particular mountain region affect the values calculated for  $\sigma_{HS}$ ? I'm guessing it wouldn't if the snow and terrain is deep but perhaps in particularly low water years it might.

The variability in summer terrain characteristics in our pooled data is rather large. Spatial average slope angles range from 4° to 60° ( $\mu$  from 0.05 to 1.22), terrain correlation lengths  $\xi$  from 6 m to 775 m and  $L/\xi$ -ratios from 3 to 40. Thus, typical summer terrain characteristics captured by coarse climate model grid cells are well represented. We now discuss this in Section 3.1.

While interannual variability in snow depth in a particular mountain region might affect the values calculated for  $\sigma_{HS}$ , when normalizing  $\sigma_{HS}$  with the corresponding spatial mean HS the regions order very closely to each other (cf. Figure 6a). This was also a finding of [1] who therefore introduced spatial mean snow depth in the parameterization for  $\sigma_{HS}$  as a climate indicator which we followed here.

While you cite Helbig et al., 2015 and Essery and Pomeroy, 2004 in the introduction there is a lot of context from those two papers which is essential to understand this current study, and I think this manuscript would benefit from including more thorough explanations and context from them. I've tried to specify several examples in the comments below which I think would help but there may be others.

Thanks for pointing that out. We address all specific comments below.

**Specific comments**

L100-106: I think the terminology "peak of winter fSCA" parametrization causes some confusion here. I recognize that  $\sigma_{HS}$  is being calibrated by mid-winter (March) data, but from your previous work (Helbig et al., 2015) you are expecting that the parametrization will apply during accumulation and melt as well. Furthermore the fSCA you describe in eq 2 is calculated from Helbig et al. (2015) assuming melt events resulting in a SCD curve. This paper never mentions the accumulation season or melt until the very end at line 445. Unless you have changed your opinion on whether or not the formulation used here and in Helbig et al., 2015 can be extrapolated outside of the peak season, I think you should mention this at some point in the introduction. If you are truly concerned that it can't be extrapolated outside of the peak-snow season I think you need to justify its potential use.

We agree this description has been confusing. The terminology "peak of winter" actually only refers to the  $\sigma_{HS}$  parameterization. We went over the manuscript to correct any further inconsistencies. By tracking snow depth over the season, a seasonal fSCA model implementation of Eq. (1) and (2) is possible. We will present this in a different article, which will be submitted soon. We rephrased the last paragraph of the conclusions to improve the seasonal fSCA outlook.

A visual representation of L,  $\xi$ , dx,dy would be helpful. E.g. representational lines on Figure 3, another panel in Figure 3, or at least explicitly refer the reader to previous work (e.g. Fig 2 of Helbig et al., 2009).

In Section 3.3 we introduce the summer terrain parameters. For a visual representation we now refer to Figure 2 of Helbig et al. (2009).

Equation 1: I think it would be helpful to state that equation 1 has been shown to reasonably parametrize fSCA for both nonmoutainous and mountainous regions, while the relationship in Eq 2 is derived using only mountain data (at approx. seasonal peak, if you'd like). And/or state this distinction in the introduction (e.g. at line 72, "While the standard deviation of snow depth introduced by Essery and Pomeroy did not depend on subgrid terrain characteristics, the formulation shown in Equation 2 was introduced by Helbig et al. (2015) in order to better model Equation 1 in mountainous terrain.

Thanks. We now point this out in Section 3.3 as well as in the introduction.

You refer the reader to [1] at the start of Section 3, but it's not clear if this is to describe the domain sampling procedure, or even if the same method used in the 2015 paper is used in this manuscript. The 2015 reference specifies 12 domain sizes between 50 and 3000m were randomly sampled. In this manuscript there are 20 bins shown on Fig 4. Please provide additional information on how each data set/scene is decomposed into domain sizes.

Thanks for pointing this out. We added more details to Section 3.1 and now show the numbers for the full range of the 41 domain sizes in Figure 4 instead of in bins. The geographic site sampling procedure used here is in regular grids (per various domain size) per geographic site.

L207: The symbol HS is being used to represent both the domain-average snow depth and the high-resolution observed snow depths at fine scale resolution (e.g. figures 5a, 6a, lines 73-100). I suggest you distinguish these uses.

Though high-resolution observed snow depths at fine scale resolution are actually also spatial mean values over a much finer sampling, we agree the symbol usage might have been a little confusing and rephrased or clarified the usage of the symbol HS where applicable.

L240: Do you sample the autocovariance in each domain 40 times? Why do you single out L=3km and then say you find inflection points for each domain size L?

We derived a total of 40 autocovariances from the available 3 km domains. Unfortunately, the description of the results on spatial autocorrelation caused some confusion. We improved Section 4.1 to make it more clear.

L253: This is the first time you use sqS, and  $\sigma_{sqS}$ . Again for context it would be good to mention that you are repeating previous analysis that established  $\mu$  and  $\xi/L$  as the most important correlates, and you are examining these two variables to compare to results from Skaugen and Melvold, which you do in the Discussion section.

All candidates for terrain parameters are introduced in the Methods Section 3.4. We extended this description to make it more clear that these parameters are later evaluated.

L267-280: While I understand the results shown in Figure 9, I couldn't understand your description of the methods used to produce them. I suggest removing/reordering the first 4 sentences from this paragraph. The discussion of domain size dependent fitting only confuses things when you then discuss the fit to the entire pooled data set. I suggest beginning the paragraph with "Fit parameters were first calibrated for the entire data pool yielding  $c = 0.6589 \ (\pm 0.0037) \ and \ d = 0.5638 \ (\pm 0.0043) \ with the 90 \ \% \ confidence \ interval... \ ... \ larger than the previously derived constants a, b in Eq. (2) (cf. Figure 9). For each stepwise domain size between 200 m to 5 km scale-dependent parameter values are also fit from the data (cf. individual colored lines in Figure 9)." At this point please provide a more complete description of the subsampling used to derive c, d for each step-wise domain size. What does 80% mean? Are the parameter values fitted from all the data within a randomly chosen domain of the appropriate size and this process is repeated 500 times? For domain sizes above 1km there are ison on with the discussion of how the parameters increase with L and the subsequent fitting of <math>c(L)$  and d(L)

We have rewritten Section 4.3 considering the suggestions and also adapted Section 5.3. We split the entire data pool in 25 sub pools for any available domain size between 200 m to 5 km (cf. Figure 4). Thereby, each sub data pool included all domains larger or equal to the corresponding domain size, i.e.  $L \ge 200$  m,  $L \ge 240$  m etc. ... From each of the 25 created sub data pools, we randomly took 500 sub samples where each sub sample comprised 80 % data of the sub data pool. Each of the sub sample per sub data pool was unique. Scale-dependent parameter values were derived for each of the 500 sub samples drawn from each of the 25 sub data pools (cf. individual colored lines in Figure 9). ... By fitting the ensemble median of all scale-dependent fit parameters (dark blue dots in Figure 9) across all domain sizes between 200 m to 5 km, we obtained scale-dependent parameters c(L) and d(L).

Fig 9: Please use a different description on the legend in place of f(L) - Eq. (3)' which can read as f(L) minus Equation (3)'.

Figure 9 is changed.

Section 4.4.4: Does the different choice of domain aspect ratio (square vs rectangular) affect the differences described in this section?

The domain aspect ratio is not important. The mean domain size L is derived from  $L_x = 500$

m and  $L_y = 1000$  m resulting in L = 750 m. The parameterization of [2] was developed for mean domain sizes L of 750 m. This means that, unlike the parameterization of [1], the parameterization of [2] wasn't developed across spatial scales. We added this to the discussion in Section 5.4.4.

L335-337: Please rephrase or clarify: "at these scale lengths." I think you are saying something like "Above scale-lengths of 200m all three effects (precip/wind/radiation-interactions) come into play, while we think there are different physical effects which establish the breaks at 20 and 60m," but please confirm. Also consider rephrasing "scale-independent parameterization", since the parametrization incorporates scale information from the sub-domain terrain parameters as well as in the constants (c(L), d(L)). Perhaps something like "The results presented here indicate that the model described by (eqs. 1 and 2) is a reasonable fSCA parametrization in mountainous terrain for spatial scales between 200m to 5km."

Yes, you are right and we rephrased the paragraph. While we got rid of "scale-independent parameterization" here, we kept it elsewhere when required to make clear that the empirical parameterization is not developed on fixed scale lengths but rather for a broad range in spatial scales.

Given that you are aiming to have this used as a fSCA parametrization in climate models which can still use grid scales as high as 50-100km please comment on the extrapolation of your results substantially beyond 5km.

You are right. We now comment on the applicability of the parameterization for larger grid sizes than 5 km in Section 5.1.

"The results presented here indicate that the model described by Eq. (1) and (3) is reliably parameterizing the spatial snow distribution shaped by the longer range precipitation, wind and radiation interactions with topography for spatial scales between 200 m and 5 km. Above the detected scale range of around 200 m not only the spatial autocorrelations approach zero (Figure 7), but normalized  $\sigma_{HS}$  clearly start levelling out as well as the normalized variability of  $\sigma_{HS}$  among similar sized L (Figure 6). Thus, even though we could not verify the fSCA parameterization for length scales larger than 5 km, we believe that as long as grid cell mean slope angles are larger than zero, Eq. (1) and (3) might also hold for larger grid cell sizes than the 5 km."

L343: Do you mean "for spatial scales between 0.5km and 1km"?

[2] investigated correlations for spatial snow depth distributions in grid cells of 0.5 km x 1 km. We rephrased this.

L357: "Furthermore, larger (about 17% and 45%, respectively) but overall consistent constant fit parameters were obtained compared to those from [1] based on a more limited number of data sets and just two geographic regions (cf. a, b...

Rephrased.

L411-413: I'd suggest that the appropriate standard for how different parametrizations perform is the range of MPE seen among different regions, not the difference between your previous calibration and the current one.

We completely agree, which is why we presented our model performances (by MPE) for the different regions (Figure 11). However, we additionally evaluated the empirical parameterization of [1] on the large evaluation data pool of the present study to investigate if the underlying functional form of our parameterization is reliable and if the parameterization works in independent geographic regions. Only one out of the 11 data sets was used for developing the empirical parameterization of [1].

**Technical comments**

L185: 3m to 5km Corrected.

Discussion, several places: "origin" as a verb - "originate" Thanks, corrected.

L379: I'd suggest splitting this sentence in two. Done, thanks.

L400: rephrase Rephrased.

L395: "decrease from 80cm..." Done.

L409: "sensitivity"? Corrected.

\*References

- N. Helbig, A. van Herwijnen, J. Magnusson, and T. Jonas. Fractional snow-covered area parameterization over complex topography. *Hydrol. Earth Syst. Sci.*, 19:1339–1351, 2015.
- [2] T. Skaugen and K. Melvold. Modeling the snow depth variability with a high-resolution lidar data set and nonlinear terrain dependency. *Water Resour. Res.*, 55:9689–9704, 2019.

---

## Author Comment (AC2) · 19 Nov 2020

We thank the reviewer# 2 for the review and the constructive comments! All reviewer comments (in italics) are addressed below.

**Specific comments**

*The 11 diverse spatial, high-resolution snow depth data sets were pooled to develop an empirical parameterization for sigma snow depth. There is some discussion of how snow depth data from the sensors/platforms affect results in different sections of the paper. Can a summary of which sensor/platform provides the "best" snow depth data set resulting in a better parametrization for sigma snow depth?*

To improve the $\sigma_{HS}$ parameterization the most accurate platform for fine-scale spatial snow depth data acquisitions is most likely airborne laser scanning (ALS). However, as outlined by [2, 1], given that ALS is still very costly airborne digital photogrammetry is an economic alternative, in particular when performed with cost-effective unmanned airborne vehicles (UAS) [1].

While we weren't able to clearly relate some of the poorer region-wise performances to uncertainties related to the platform, other studies entirely focused on performing extensive inter-comparisons between these methods for large-scale snow depth mapping in alpine terrain. We now refer the reader to these studies in Section 5.4.3.

*Done with snow depth data sets at annual maximum snow cover, how might parameterization of sigma snow depth with data sets collected at mid-season or late season of snow cover affect the results? Is there a preferred time in relation to seasonal snow cover to collect a high-resolution snow depth dataset? Is it possible to use multiple snow cover data sets collected at a site at different times during the season to parameterize sigma snow depth?*

By parameterizing $\sigma_{HS}$ using peak of winter snow depth data gathered in mountainous terrain, we derived a formulation of the spatial snow depth distribution at peak of winter. Since we did not have available similar detailed data sets during accumulation and melt, we did not parameterize $\sigma_{HS}$ during other periods than around approximate time of peak of winter. However, since $fSCA$ is a crucial model parameter, our overall goal is to describe the $fSCA$ curve throughout a snow season. For this we needed a reliable scale-independent peak of winter $\sigma_{HS}$ parameterization for mountainous terrain. Soon, we will submit our work on such a seasonal $fSCA$ algorithm. When using current snow depth in the peak of winter $\sigma_{HS}$ parameterization large errors in modelled $fSCA$ resulted for instance during the ablation period. However, by tracking snow depth over the season and by accounting for alternating snow accumulation and melt events during the season, these errors decreased considerably and seasonal trends of $fSCA$ were overall well parameterized.

We rephrased the last paragraph of the conclusions to improve the seasonal $fSCA$ outlook.

**Technical comments**

*Line 108: Suggest delete "large quantity", it is an unnecessary qualitative description of*

*data used.*

Done.

*Line 137: The words "than for the" cause some confusion. Was the ALS data processed similar to the ASO campaigns or different from those campaigns?*

The ALS data was processed similar to the ASO campaigns. We rephrased that.

*Line 173: "lower zero" should be lower than 0, or snow depth $\leq$. "above" could be changed to ¿ . And units should be given with "threshold of zero"*

Changed.

*Line 228: "pearson" should be capitalized, prop noun. Applies throughout the paper.*

Changed.

*Line 307: The expression "strike out" would be better stated as standout.*

Thanks.

*Line 398: Please clarify "not origin". Possibly originate is a word that could clarify source.*

You are right- we changed that.

*References

[1] Y. Bühler, M. S. Adams, R. Bösch, and A. Stoffel. Mapping snow depth in alpine terrain with unmanned aerial systems (UASs): potential and limitations. *The Cryosphere*, 10(3):1075–1088, 2016.

[2] Y. Bühler, M. Marty, L. Egli, J. Veitinger, T. Jonas, P. Thee, and C. Ginzler. Snow depth mapping in high-alpine catchments using digital photogrammetry. *Cryosphere*, 9:229–243, 2015.

---

## Author Response (AR2)

We thank the editor for the review and the constructive comments! All editor comments (in italics) are addressed below.

*Line 30: "fSCA is also parameter in hydrological models to scale water discharges in the different model grid cells managing in this way appropriately basins water supply" – add "a" between "also" and "parameter"; the second half of the sentence "in the different model grid cells managing in this way appropriately basins water supply" is unclear as written. Maybe something like "scale water discharges appropriately to help manage basins water supply"?*

Thanks. Changed.

*Line 79: "Snow depth data at these high resolutions now allow to statistically analyze spatial snow depth patterns for various purposes" – missing a word between "allow" and "to"*

Rewritten.

*Line 381: Change "(about 17 % respectively 45 % larger)" to "(about 17% and 45%, respectively)"*

Changed.

*Line 403: Similarly, change "12 % respectively 38 %" to "12% and 38%, respectively"*

Changed.

*Both reviewers indicated that it was unclear from the text whether a sigma snow depth parameter derived at peak of winter could be used to develop an fSCA curve throughout the season. The text is improved with the removal of "peak of winter" to describe fSCA and in the conclusion (lines 475-487), but I suggest additional explanation in the introduction may help reduce any confusion as readers go through the paper (e.g. at line 74, at the end of the paragraph where the parameter is introduced).*

We added additional information on a seasonal model implementation at the end of the introduction.

*The summer terrain parameters are defined in section 3.3, but a discussion on the variability of those characteristics in the pooled data is given in section 3.1. Is it possible to rearrange the text so the parameter definition comes first?*

We agree and put Section 3.3 at the beginning of the Methods section.